# Observation of Neutron-Induced Absorption and Phase-Mismatch for Quasi-Phase-Matched Second Harmonic Generation in Congruent Lithium Niobate

An-Chung Chiang [1,*], Jiunn-Hsing Chao [1], Shou-Tai Lin [2] and Yuan-Yao Lin [3]

1   Nuclear Science and Technology Development Center, National Tsing-Hua University, Hsinchu 300044, Taiwan; jhchao@mx.nthu.edu.tw
2   Department of Photonics, Feng Chia University, Taichung 407802, Taiwan; stailin@mail.fcu.edu.tw
3   Department of Photonics, National Sun Yat-sen University, Kaohsiung 804201, Taiwan; yuyalin@mail.nsysu.edu.tw
*   Correspondence: acchiang@mx.nthu.edu.tw

**Abstract:** We report an experimental observation of radiation-induced optical absorption in undoped congruent lithium niobate ($LiNbO_3$) crystals between the 190 nm and 3200 nm wavelength range. It was found that high-dose (up to 288 Gy) gamma rays did not significantly affect the optical absorption of $LiNbO_3$ during that wavelength range. However, upon the order of $10^{16}$ $cm^{-2}$ neutron irradiation, the change in the absorption coefficient was up to 0.6 $cm^{-1}$ at a wavelength of 532 nm and remained after 13 days in the dark at room temperature. The nonlinear optical performance was characterized by conducting third-order quasi-phase-matched second harmonic generation in neutron-irradiated periodically-poled $LiNbO_3$, showing that the phase-matching condition was altered, and the conversion efficiency was still comparable with a non-irradiated one.

**Keywords:** lithium niobate; neutron irradiation; quasi-phase-matching; PPLN





## 1. Introduction

The quasi-phase-matching (QPM) technique has drawn much attention due to its promising applications in various areas in the field of photonics, such as nonlinear frequency conversion and electro-optic modulation [1–3]. New solid-state laser light sources employing QPM devices have also been developed in recent years [4]. Lithium niobate ($LiNbO_3$) covers a useful transparency range from the visible to infrared region and is widely available in large homogeneous crystals. In addition, $LiNbO_3$ has a comparably large effective nonlinear coefficient, thus, is advantageous to many nonlinear optical applications. The domain-inverted $LiNbO_3$, also known as periodically poled lithium niobate (PPLN), has become an important QPM device in nonlinear frequency mixing processes due to its high nonlinearity, artificial phase-matching condition, and repeatable fabrication [5,6]. The fabricating techniques are now well developed and device-quality PPLN crystals can be routinely manufactured.

For many years, PPLN devices were extensively used for laser systems and integrated photonic devices. Increasing interests have also been devoted to the applications of photonics technologies to space, including high-speed light modulation, laser remote sensing, and so on. It is believed that PPLN devices should be one of the promising materials for the application of photonics devices that can be applied in harsh radiation environments such as space, aviation, high-energy accelerator facilities, and nuclear reactors. The interest in the behavior of $LiNbO_3$-based devices close to the core of a nuclear reactor started in the 1970s [7]. Radiation damage to $LiNbO_3$ upon neutron irradiation was found to be significant with neutron fluences up to $10^{20}$ $cm^{-2}$ and the loss of optical birefringence and piezoelectric response was demonstrated [8]. For neutron fluences up to $4 \times 10^{17}$ $cm^{-2}$,

there is an expansion along the c-axis and a shift towards longer wavelengths of the UV absorption edge [7]. Irradiation of lower neutron fluences shows that the refractive index of LiNbO$_3$ changes with the neutron fluences [9,10]. Although previous studies provide little detail on the actual neutron spectrum, intermediate fluences ($10^{15}$~$10^{16}$ cm$^{-2}$) of neutron irradiation alter some optical properties of LiNbO$_3$ without massively destroying the crystal and causing the loss of other advantageous characteristics. Studies concerning the intermediate-fluence radiation effects on LiNbO$_3$ have been concentrated on the crystal properties themselves. The available experimental data focusing on the nonlinear optical performance of PPLN devices are relatively rare. Tsing Hua Open-pool Reactor (THOR) at National Tsing-Hua University (NTHU), Taiwan, is a 2-MW light-water nuclear reactor for research in Taiwan. The vertical tubes near the edge of the reactor core of THOR provide a neutron fluence rate of up to ~$10^{12}$ cm$^{-2}$s$^{-1}$, which are suitable neutron sources for examining the nonlinear optical performance of PPLN devices upon intermediate-fluence neutron irradiation.

In this paper, we utilized the neutron source at THOR to irradiate several congruent LiNbO$_3$ samples (including unpoled LiNbO$_3$ and PPLN crystals) and conducted third-order QPM second harmonic generation (SHG) experiments. A 1064 nm laser was used as the fundamental wave to examine the change in the phase-matching condition of SHG and its efficiencies after neutron irradiation. Though THOR provides fission neutron irradiation accompanied by gamma irradiation, additional test samples were also independently irradiated by a Cobalt 60 ($^{60}$Co) gamma-ray source at NTHU for comparison.

## 2. Experimental Methods and Results

We prepared four unpoled x-cut LiNbO$_3$ crystals (namely sample #0, #1, #2, #3) for the test of optical transmission for different irradiation conditions. The dimension of the unpoled x-cut LiNbO$_3$ crystals are 12 mm (y) × 8 mm (z) × 1 mm (x). The surfaces are un-coated. To test the nonlinear optical performance, we also prepared four PPLN samples (namely, sample #0A, #4, #5, #6) for irradiation. The four PPLN samples were cut from the same LiNbO$_3$ wafer after the poling process was completed and had the same QPM grating period of 20 μm and the identical dimension of 15 mm (length) × 2 mm (width) × 0.5 mm (thickness). This ensured that the optical properties of the four samples were identical. The 20-μm QPM period accounts for 1064 nm pumped third-order SHG with a phase-matching temperature of 102.5 °C. All the PPLN samples have anti-reflection coatings on both their end surfaces at 1064 nm, with a transmission of 99.6%.

The neutron irradiation experiments of the test samples were carried out at THOR, which is a 2-MW light-water nuclear reactor for research in Taiwan. The test samples were irradiated at the vertical tube, namely VT-4, of THOR. The vertical tube is located at the edge of the reactor core, corresponding to approximate thermal and fast neutron with a ratio of thermal-to-fast neutron around 3~5 [11]. Details on the conditions of the neutron irradiation by the THOR neutron source are listed in Table 1. The fast and thermal neutron fluences were determined according to the gold foil activation experiments, as reported in ref. [11]. Test sample #0 and #0A, as reference, were not irradiated.

**Table 1.** Neutron/gamma irradiation conditions of the lithium niobate test samples.

| Test Samples | | P (MW) | $t_{irrad}$ (h) | $\Phi_t$ (#/cm$^2$) | $\Phi_f$ (#/cm$^2$) | $D_g$ (Gy) |
|---|---|---|---|---|---|---|
| #0 LN | #0A PPLN | - | - | - | - | - |
| #1 LN | #4 PPLN | - | 350 | - | - | 288 |
| #2 LN | #5 PPLN | 1.5 | 1 | $4.4 \times 10^{15}$ | $1.3 \times 10^{15}$ | 144 |
| #3 LN | #6 PPLN | 1.5 | 2 | $8.8 \times 10^{15}$ | $2.6 \times 10^{15}$ | 288 |

P: reactor power; $t_{irrad}$: neutron irradiation time; $\Phi_t$: thermal neutron fluence; $\Phi_f$: fast neutron fluence; $D_g$: gamma dose. LN: LiNbO$_3$.

Since the irradiation experiments were performed near the fission reactor core, accompanying the emission of gamma rays from fission products with the neutron irradiation is

expected. According to a calibration experiment in which a thermoluminescence dosimeter was used, the dose rate of gamma rays near the THOR irradiation tube was found to be about 0.04 Gy/s. We used the $^{60}$Co gamma-ray source to independently irradiate samples #1 and #4 to obtain gamma-only results. The accumulated gamma-ray dose was controlled to be the same as that being irradiated in the nuclear reactor for 2 h. Using the current $^{60}$Co source, it took 350 h to reach the same gamma-ray dose.

It is known that the irradiation of neutrons and gamma rays would change the refractive index of LiNbO$_3$ [9,10], so as to alter the phase-matching condition of the nonlinear optical process within it. It is also necessary to know the change in optical transmission after neutron and gamma-ray irradiation, since the nonlinear optical performance is strongly affected by the intensities of interacting waves. Samples #0, #1, #2, and #3 are x-cut LiNbO$_3$ crystals for measuring transmission spectra. The transmission spectra of the irradiated samples were measured by a Shimadzu grating spectrophotometer (UV-3101PC). The measurable wavelength range is from 190 nm to 3200 nm. The light sources are switched automatically in conjunction with wavelength scanning. To efficiently measure the full wavelength range, we initialized the configuration to have a scanning speed of 100 nm/min with a 2 nm sampling interval. Some unexpected spikes near the 850 nm wavelength in the measured transmission spectra during the switching of light sources occurred and were neglected for the broad range measurement.

### 2.1. Gamma Energy Spectrum of Neutron Activation Products

In order to see neutron-activated products in neutron-irradiated LiNbO$_3$, sample #3 was sent to a gamma energy spectrum analyzer after irradiation. After 24 h of cooling, the sample was measured using a high-purity germanium detector (GC3020, Canberra Industries, Inc., Meriden, CT, USA) coupled with a multichannel analyzer and a software package (Genie 2000, Canberra Industries, Inc., Meriden, CT, USA). The measured gamma-ray spectrum of the chip irradiated by neutrons is shown in Figure 1. The prominent gamma rays (935 keV) were emitted from $^{92m}$Nb (half-life 10.15 d), which is produced through $^{93}$Nb (n,2n) $^{92m}$Nb reaction. The measured activity of $^{92m}$Nb was about 2000 Bq. Some interfering gamma rays emitted from $^{24}$Na, $^{82}$Br, $^{122}$Sb, etc. were also observed. Those nuclides were formed by neutron capture reactions with impurities in the sample. The LiNbO$_3$ sample is optical-grade and contains a very small number of impurities. The remaining activity was insignificant after cooling. The radiation dose rate on the sample surface after 24 h of cooling was only 0.5 μSv/h, which mean that it was not necessary to use a lead container for radiation shielding while retrieving the sample from the neutron irradiation facility. Seven days after the first gamma spectrum measurement, the sample was measured again. It was found that most of the short-lived radio nuclides had decayed. $^{92m}$Nb can still be measured, but the activity is significantly reduced in accordance with its 10.15-day half-life. In-crystal gamma emission should not affect the following experiments.

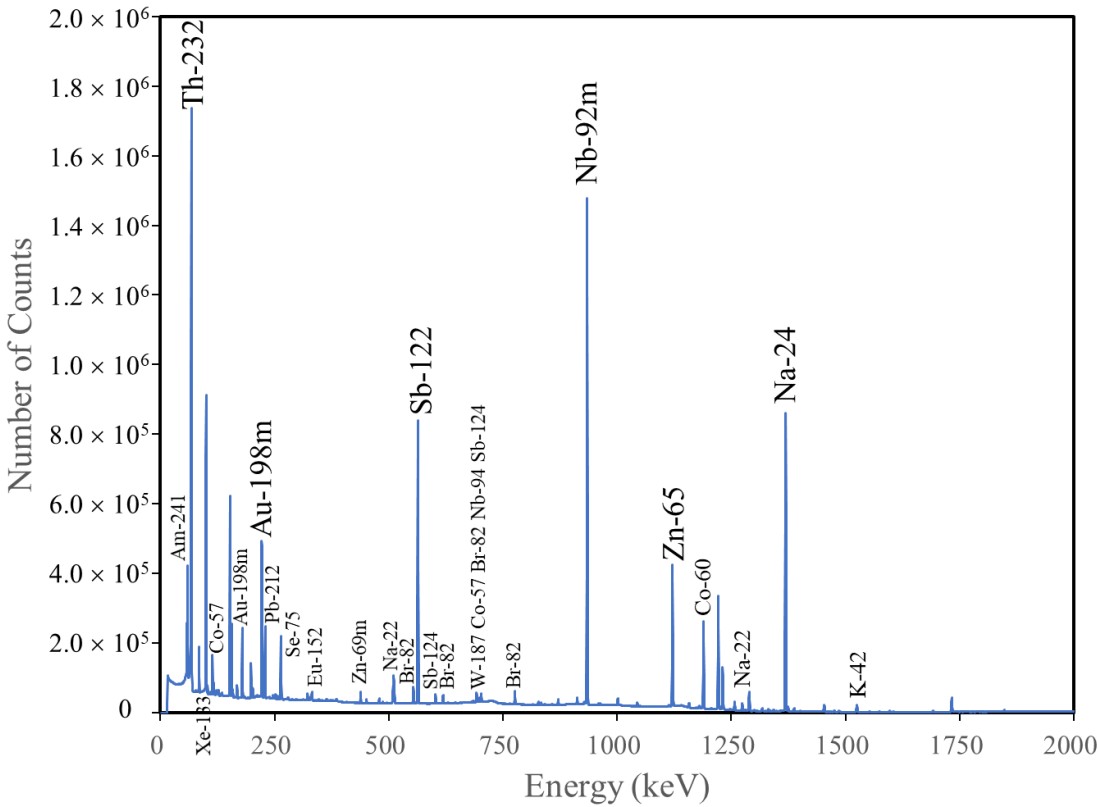

**Figure 1.** Gamma energy spectrum of a neutron-irradiated LiNbO$_3$ sample. A LiNbO$_3$ sample (12 mm × 8 mm × 1 mm; 0.45 g) was prepared and irradiated by neutrons for 2 h by THOR, where neutron fluence rate is about $10^{12}$ cm$^{-2}$s$^{-1}$. After 24 h of cooling, the sample was measured using a high-purity germanium detector coupled with a multichannel analyzer and a software package.

*2.2. Change in Absorption after Irradiation*

Figure 2 shows the absorption coefficient change for neutron and gamma-irradiated LiNbO$_3$ as a function of wavelength. The absorption coefficient change, $\Delta\alpha$, is the relative change in absorption coefficient with respect to that of sample #0 and is defined by

$$\Delta\alpha_{\#1,2,3} = -\frac{1}{L}\ln\left(\frac{T_{\#1,2,3}}{T_{\#0}}\right)$$

where $L$ is the thickness of the sample, $T_{\#0}$ is the original transmission of sample #0, and $T_{\#1,2,3}$ are the original transmission of samples #1,2,3. The change in absorption coefficient for the gamma-irradiated sample (#1) is very small, showing that congruent LiNbO$_3$ is less susceptible to gamma irradiation in optical transmission during the measured wavelength region. However, the neutron-irradiated samples (#2 and #3) showed a significant change in optical absorption, especially during the visible region. From the actual photo of the four samples, the neutron-irradiated samples are apparently "brown", demonstrating the increased absorption in the shorter visible region. This broad band absorption in the visible region is primarily due to the oxygen vacancies induced by radiation, as the displacement damage is related to the oxygen vacancies and their corresponding interstitials trapped within the lattice [12–14]. The radiation-induced change in absorption was not smoothly changed with the wavelength in Figure 2. This was due to changes in the light sources in the spectrophotometer and the etalon effect of the 1-mm-thick sample. Since sharply decreased transmission occurs in the cut-off wavelength region (200~300 nm), the deduced radiation-induced changes in absorption during that region was very sensitive to tiny measurement instabilities and could be ignored.

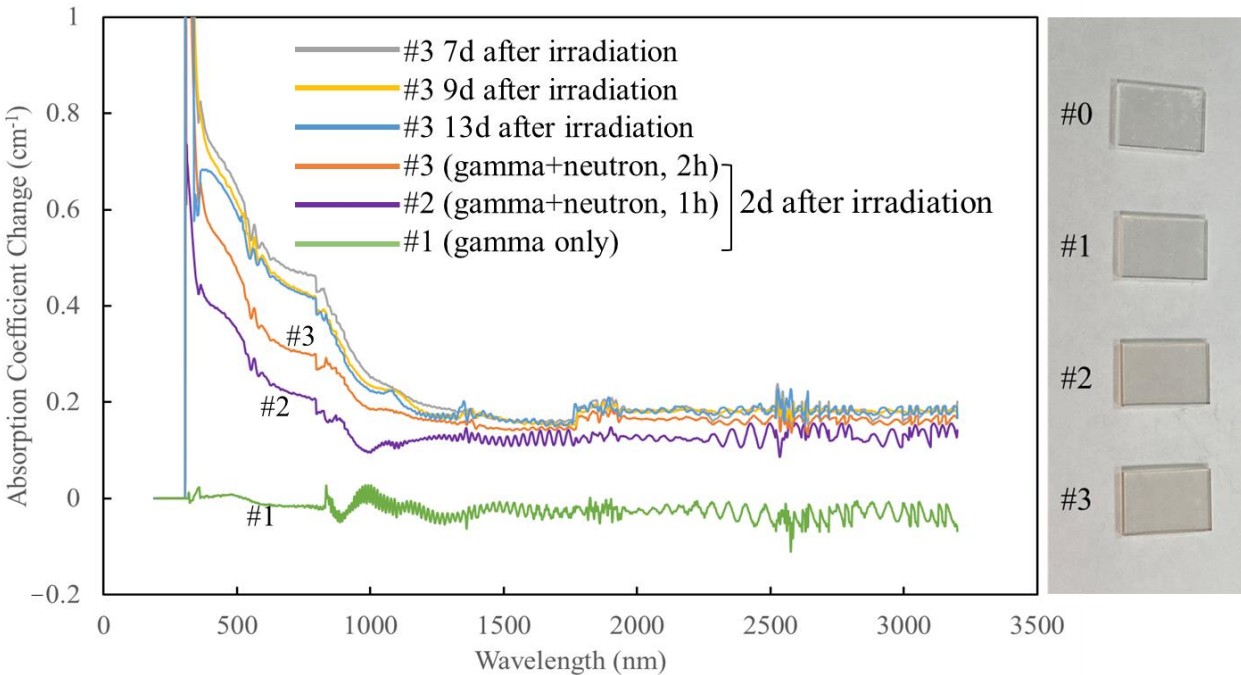

**Figure 2.** Absorption coefficient change for neutron and gamma-irradiated LiNbO₃ as a function of wavelength. The data curves in the figure are the absorption coefficient changes relative to the reference sample #0. Sample #1 was irradiated with 288-Gy gamma rays and its absorption coefficient changes by almost zero. Samples #2 and #3 were irradiated with neutrons and showed significant changes in absorption. The absorption coefficient change was continuously occurring after the neutron-gamma irradiation. The right photo shows the appearance of the four samples 13 days after irradiation.

Figure 3 shows the room-temperature dark change in absorption (for sample #3) as a function of days after neutron irradiation for four specific wavelengths (532 nm, 1064 nm, 1550 nm, and 3200 nm). At room temperature, the absorption continuously increased from day 2 to day 7 after irradiation, and then slightly decreased from day 7 to day 9, ending stationary after day 13. The change in absorption after neutron irradiation was more significant for the 532 nm wavelength and was much smaller for the 1064 nm, 1550 nm, and 3200 nm wavelengths. The dark change in absorption was not obvious for 1550 nm.

Thermal neutrons produce displacement damage mostly through (n, γ) reactions, while fast neutrons produce displacement damage through elastic collisions due to the recoil induced by the emission of the gamma photons [15,16]. The $^6$Li (n, α) $^3$H nuclear reaction also needs to be considered for LiNbO₃ since α and $^3$H also produce displacement damage [17]. In fact, the $^6$Li (n, α) $^3$H reaction is the main source of damage, since the thermal cross section for this reaction is larger than the absorption cross sections of all other constituents. In addition, the damage production due to the charged MeV reaction products is high [17]. The displacement damage alters the optical characteristics of LiNbO₃ and continuously relocates after neutron irradiation under room temperature until the whole structure becomes stable several days after irradiation.

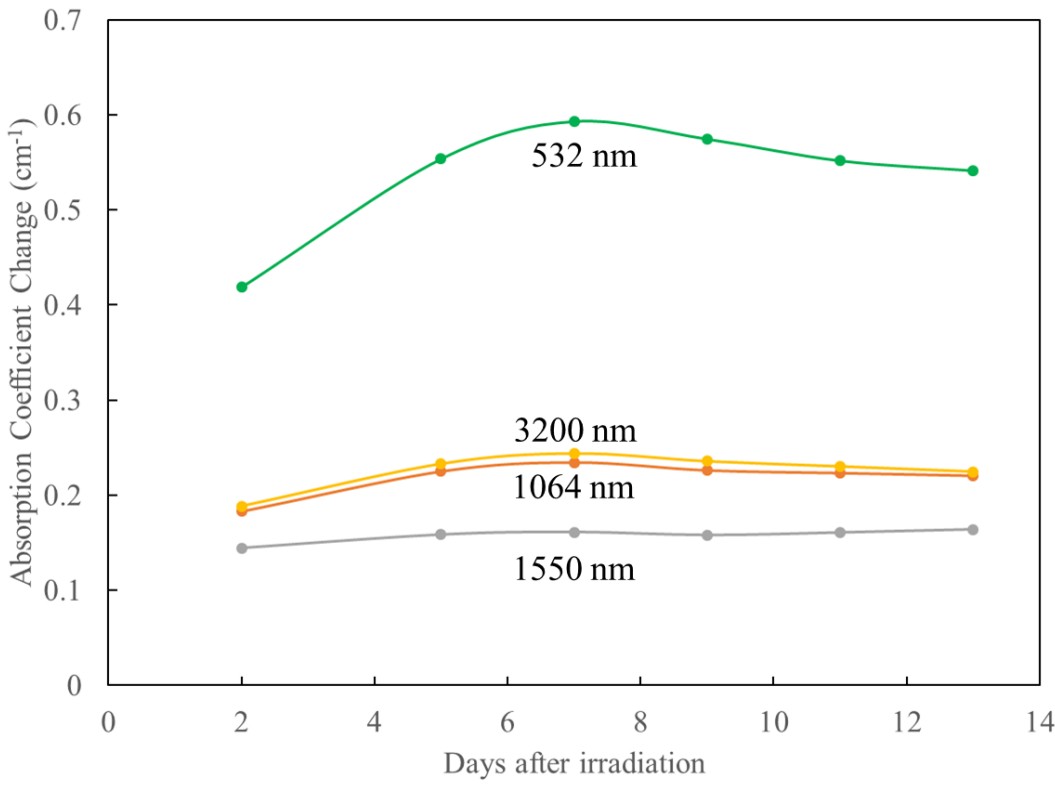

**Figure 3.** Dark change in absorption (for sample #3) as a function of days after neutron-gamma irradiation for four specific wavelengths. The absorption increased from day 2 to day 7 after irradiation and slightly decreased from day 7 to day 9. The change in absorption reached stationary since day 13. The dark change in absorption is not obvious for 1550 nm.

### 2.3. QPM SHG Performance after Irradiation

After radiation-induced absorption was confirmed, we conducted a third-order QPM SHG experiment for radiation-irradiated PPLN samples to examine their nonlinear optical performance. The schematic experimental setup is shown in Figure 4. A pulsed laser was used as the fundamental source. It is a 1064 nm passively Q-switched Nd:YAG laser, generating 1-ns pulses with a pulse energy of 9 μJ and a 3.76-kHz repetition rate. The peak power is 9 kW and the average power is 34 mW. An attenuator-isolator set, consisting of a Faraday rotator following a half-wave plate, controls the pump energy and eliminates the optical feedback. The focusing mirror focuses the pump laser beam to the PPLN crystal. The oven, containing the PPLN crystal, maintains the temperature at the SHG phase-matching temperature. The SHG temperature tuning curve acts as a sinc function and can be measured by recording the SHG power while tuning the PPLN temperature. The phase-mismatch caused by neutron irradiation can thus be determined in terms of the changes in the phase-matching temperature.

Due to radiation-induced absorption, it was expected that the SHG conversion efficiency would be decreased, as would the effective nonlinear coefficient after irradiation. However, the high-peak power of the fundamental laser would still result in high-efficiency SHG, even if significant radiation-induced absorption exists. Thus, the focusing condition was intentionally not optimized and the conversion efficiency was less than 10% to avoid running into a highly depleted regime. The SHG power at 532 nm was measured by a calibrated thermal detector. The response time of the thermal detector eliminated the effect of the ~10% Q-switch energy jitter and precisely measured the average SHG power.

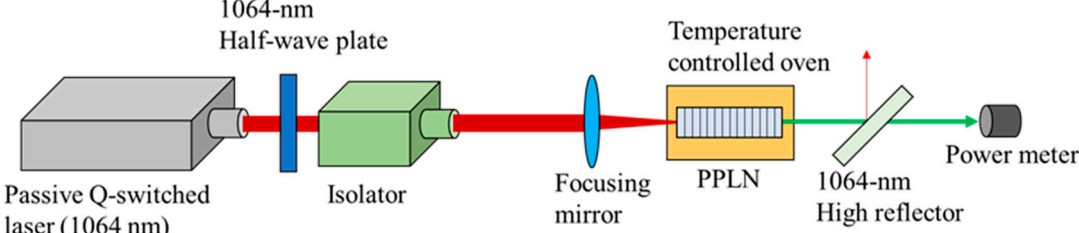

**Figure 4.** The schematic experimental setup of third-order SHG in radiation-irradiated PPLN. A 1064 nm high reflector was used to remove the un-depleted 1064 nm laser power. The PPLN was in a temperature-controlled oven. A power meter was used to measure the 532 nm laser power. Temperature tuning curves were recorded.

Owing to the dark recovery observation, we conducted the SHG experiments 13 days after irradiation (at the 14-th day). Figure 5 shows the phase-matching temperature tuning curve obtained for the neutron-irradiated PPLN samples. Sample #0A was a reference PPLN crystal without any irradiation. Samples #5 and #6 were irradiated by THOR for 1 h and 2 h, respectively. The gamma-irradiated PPLN crystal (sample #4) had a similar temperature tuning curve as sample #0A and was not shown in Figure 5 for a clearer presentation. Notice that the discrepancy between the measured phase-matching temperature and the theoretical value for sample #0A is due to the temperature gradient between the temperature sensor and the PPLN crystal and the possible fabrication error of the PPLN gratings.

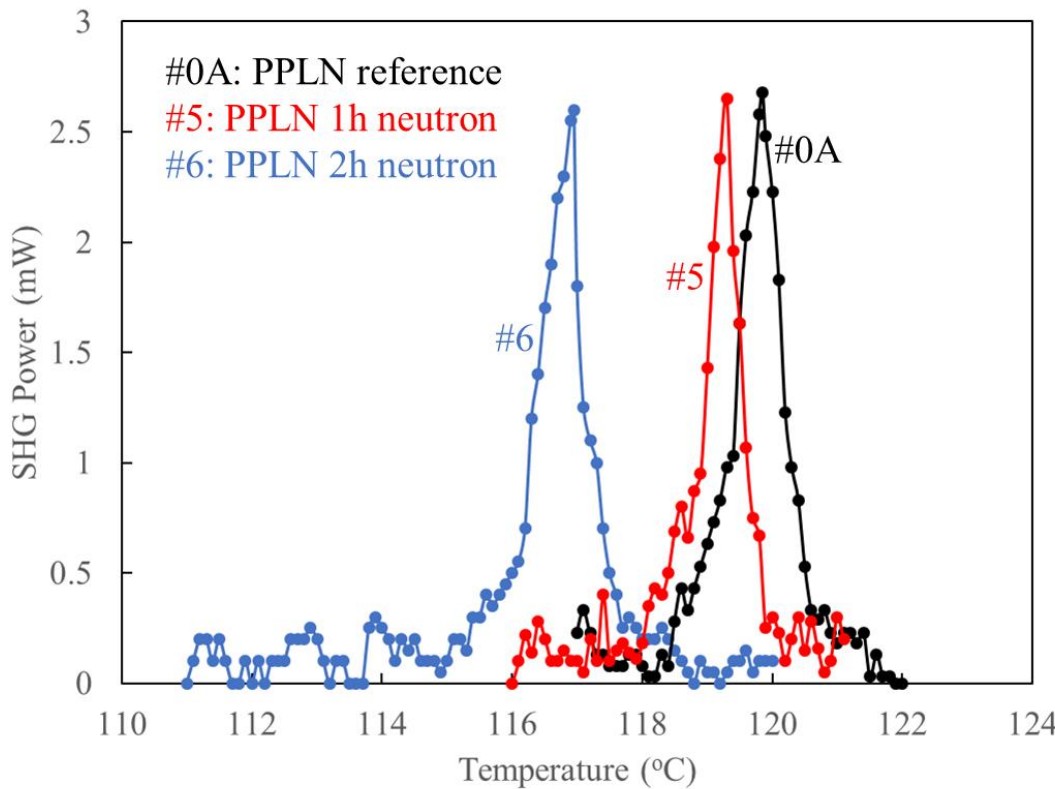

**Figure 5.** Phase-matching temperature tuning curve obtained for neutron-irradiated PPLN. Sample #0A is a reference PPLN crystal without any irradiation. Sample #5 and #6 are irradiation by THOR for 1 h and 2 h, respectively. The $^{60}$Co gamma-irradiated sample (#4) has a similar temperature tuning curve with respect to sample #0A and is not shown in this figure.

The experimental results showed that gamma rays did not affect the optical absorption of LiNbO$_3$ with the radiation dose up to 288 Gy. The SHG phase-matching condition was

also not altered. However, the irradiation of neutrons caused strong absorption during some specific wavelength regions, especially in the visible and the mid-infrared region. According to the Kramers–Krönig relations in nonlinear optics, the strong absorption should result in the dispersion of optical materials. Nonlinear optical processes in a neutron-irradiated LiNbO$_3$ crystal would have a different phase-matching condition. Figure 5 demonstrates that the phase-matching temperature was down-shifted by ~1 °C and ~3 °C for the 1-h- and 2-h-neutron-irradiated PPLN samples, respectively. Additionally, as shown in Figure 5, the SHG conversion efficiencies for the two neutron-irradiated PPLN samples were comparable to that of the non-irradiated one. It is known that the absorption loss of the interacting waves dramatically decreases the conversion efficiency of a second-order nonlinear process. In our experiment, the increased absorption coefficient (~0.55 cm$^{-1}$) for 532 nm corresponded to a ~50% transmission loss for the 1.5-cm-long PPLN crystal. For the neutron-irradiated PPLN crystals, we expected to obtain SHG efficiencies which are significantly lower than the non-irradiated one. However, we found that the SHG efficiencies are still comparable to the non-irradiated one. Even though the measured SHG efficiencies for the two irradiated samples seem to be only slightly smaller, the result still demonstrated that the nonlinearity of LiNbO$_3$ after irradiation would be increased. It is evident that neutron irradiation not only introduced optical absorption but also caused increased effective nonlinearity in LiNbO$_3$ crystals. Further investigation needs to be carried out for this observation.

Photorefractive distributed feedback (DFB) grating in PPLN is reported in our previous experiment, in which the preliminarily results of DFB optical parametric oscillation (DFB OPO) in PPLN were demonstrated [18]. During the experiment in ref. [18], we found that UV-induced infrared absorption (UVIIRA) is an issue for laser application. Investigations on UVIIRA have been conducted and reported [19]. Ultra-long lifetime UVIIRA in congruent LiNbO$_3$ was observed, showing that absorption coefficient change in congruent LiNbO$_3$ is significant if UV light is used for creating photorefractive grating in a LiNbO$_3$ crystal. This makes UV photorefractive scheme inefficient for DFB OPO. Irradiation of gamma rays with a suitable dose would greatly introduce index change in bulk LiNbO$_3$ crystals [9] without affecting optical absorption and nonlinearity. This could be an alternative way of producing DFB grating for DFB OPO if an appropriate method of spatially distribution of gamma rays is utilized; for example, the periodic deposition of lead upon the surface of PPLN. Neutrons could be also useful for creating non-volatile index changes in bulk LiNbO$_3$ crystals, despite the disadvantage of introducing optical absorption. According to our experimental results, the nonlinear optical performance in neutron-irradiated PPLN crystals for third-order QPM SHG was not susceptible after neutron irradiation. Irradiation of neutrons could also be useful for nonlinear optical applications that require non-volatile changes in the refractive index for phase adjusting or modulation.

## 3. Conclusions

In conclusion, we have demonstrated the observation of radiation-induced absorption and phase-mismatch for third-order QPM SHG in congruent LiNbO$_3$ crystals. It was found that gamma rays up to 288 Gy did not affect the optical absorption and the phase-matching of QPM SHG in LiNbO$_3$. On the other hand, with a ~10$^{16}$-cm$^{-2}$ neutron fluence, the induced optical absorption was up to 0.6 cm$^{-1}$ for 532 nm and 0.2 cm$^{-1}$ for 1064 nm. Observations of dark recovery for optical absorption showed 13-day recovery time at room temperature. The 2-h-neutron-irradiated PPLN crystal has a lower phase-matching temperature for third-order QPM SHG. The increase in nonlinearity after neutron irradiation was observed and seemed to compensate for the radiation-induced optical absorption, which would dramatically decrease the SHG efficiency. LiNbO$_3$ devices with photonic structures such as waveguides or gratings can be potentially engineered by the use of gamma rays or neutrons. In addition, the radiation-induced optical absorption in LiNbO$_3$ crystals and phase-mismatching in PPLN crystals can be an indication of the dose of neutron irradiation.

An alternative method could be to measure and calibrate the neutron fluence inside a nuclear reactor or for other neutron sources.

**Author Contributions:** Conceptualization, A.-C.C.; methodology, A.-C.C.; software, Y.-Y.L.; validation, A.-C.C., J.-H.C. and S.-T.L.; formal analysis, A.-C.C. and Y.-Y.L.; investigation, A.-C.C. and S.-T.L.; resources, A.-C.C.; data curation, A.-C.C. and J.-H.C.; writing—original draft preparation, A.-C.C.; writing—review and editing, A.-C.C.; visualization, A.-C.C.; supervision, A.-C.C. and J.-H.C.; project administration, A.-C.C.; funding acquisition, A.-C.C. All authors have read and agreed to the published version of the manuscript.

**Funding:** This research was funded by Institute of Nuclear Energy Research (INER), Atomic Energy Council (AEC), Executive Yuan, Taiwan, Grant number NL1100468; National Tsing-Hua University (NTHU) of Taiwan, Grant number 110H0000L1.

**Institutional Review Board Statement:** Not applicable.

**Informed Consent Statement:** Not applicable.

**Data Availability Statement:** Not applicable.

**Acknowledgments:** The authors thank A. Kopeykin for measuring the transmission spectra and HCPhotonics for providing congruent $LiNbO_3$ test samples.

**Conflicts of Interest:** The authors declare no conflict of interest.

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
