# Peer review of "Observation of Neutron-Induced Absorption and Phase-Mismatch for Quasi-Phase-Matched Second Harmonic Generation in Congruent Lithium Niobate"

_photonics, doi:10.3390/photonics9040225_

Round 1

Reviewer 1 Report

The paper describes very interesting results on the SHG generation in lithium niobate.

I recommend the publication of this paper after some minor revisions.

The English will need also some minor revision although the text is clear and understandable at all instances.

Detailed comments:

1) Lines 98-99 (“It is known that the irradiation of neutrons and gamma rays would change the refractive index …”)

Please give references – even if they are repeated

2) Lines 118 – 136 and Figure 1 (radiation protection issues)

I appreciate that you were considering radiation protection issues raised by the irradiation of the lithium niobates samples.

The gamma spectroscopy measurements are well done and described but the Figure 1 is unfortunately of very poor quality. Please try to improve that or restrict yourself to the description in text.

Actually the most important issue is the production of tritium through the Li-6(n,alpha)H-3 reaction. Tritium has a half-life of about 10 years and can outgas when you heat the sample and contaminate equipment and laboratories. Fortunately, as low-energy pure beta emitter tritium is also a low risk radionuclide.

3) Lines 143 – 159 and Figure 2 (“All the three samples (#1, #2, #3) showed significant 145 change in absorption with respect to the non-irradiated one (#0)”.

Unfortunately, Figure 2 does not contain a spectrum of sample #0 for the reader to verify your statement. Please add this or if it is too close to #1 sample to be visible mention this in the figure caption and/or the text.

You should also change the line color of #3 or #2 1h since they are difficult to distinguish.

4) Lines 173 – 177 (Damage production)

“The 6Li (n, alpha) 3H nuclear reaction also needs to be considered for LiNbO3 since alpha and 3H also produce displacement damage”

This is actually the main source of damage since a) the thermal cross section for this reaction is larger than the absorption cross sections of all other constituents and b) the damage production due to the charged MeV reaction products is high.

You may also consider recoils produced by elastic scattering of the fast neutrons by the constituents (see your Ref. 12).

5) Lines 243 – 245 (“The non-central symmetry of LiNbO3 could be further increased due to the neutron irradiation so as to increase the second order nonlinearity.”)

I cannot see how the non-central symmetry should increase throughout the crystal. This would require a collective rearrangement of the Li and Nb along the c-axis.

I would rather suggest that the change in the phase-matching temperature is related to a stoichiometry change due to the irradiation (The phase-matching temperature is very sensitive to stoichiometry – see e.g. Schmidt et al., Appl. Phys. Lett. 58 (1991) 34)

A stoichiometry change due to the “destruction” of Li-6 by the neutron irradiation probably cannot explain this change even if one assumes 100% of the neutrons are initiating this process.

On the other hand, Li2O out-diffusion has been observed in Lithium niobate damaged by ion implantation even at room temperature and might also occur here.

6) Lines 259 – 261 (“Neutron could be also useful for creating nonvolatile index changes in bulk LiNbO3 crystals, except the disadvantage of introducing optical absorption.”)

The disadvantage of optical absorption could be removed by low-temperature annealing at 300 ºC (e.g. Marques et al. Nucl. Instr. Meth.  B141 (1998) 326) which even seems to increase the actual structural damage.

This might be a future work – but take care to manage the tritium release during annealing.   

7) Suggestions (optional):

  • Use the more correct term “fluence” instead of “flux” in the paper.
  • Use the SI unit for dose “Gy” instead of “krad” (1 krad = 100 Gy)  

Reviewer 2 Report

The authors have demonstrated the observation of radiation induced absorp- 267 tion and phase-mismatch for third-order QPM SHG in congruent LiNbO3 crystals.  The reviewer recommends that the paper can be published in the journal Photonics after a minor revision revision. The following are the reviewer's comments and suggestions to enhance this manuscript:

(1) The icon font size of figure1 should be adjusted bigger to make it more comfortable. Fig. 1 left ordinate missing unit (should be a. u.)

(2)The author should change the yellow colour font in 7th page of manuscript.

Reviewer 3 Report

This is an interesting article that can be recommended for publication, but only after an interpretive analysis of the absorption spectra shown in Fig. 2.  It seems that the authors did not fully study the state of the problem and did not familiarize themselves with what has already been done in the field of radiation defects. I have to recommend to authors to read the following papers:

Hodgson, E. R., & Agullo-Lopez, F. (1987). Oxygen vacancy centres induced by electron irradiation in LiNbO3. Solid state communications64(6), 965-968.

Hodgson, E. R., & Agullo-Lopez, F. (1989). High-energy electron irradiation of stoichiometric LiNbO3. Journal of Physics: Condensed Matter1(50), 10015.

Hodgson, E. R., & Agullo-Lopez, F. (1991). Colouring and annealing behaviour of electron irradiated LiNbO3: Fe. Journal of Physics: Condensed Matter3(3), 285.

Potera, P., Ubizskii, S., Sugak, D., & Lukasiewicz, T. (2007). Colour centres in LiNbO3: Fe and LiNbO3: Cu crystals irradiated by 12C ions. Radiation measurements42(2), 232-235.

and references therein.

Round 2

Reviewer 3 Report

the authors have significantly improved the manuscript, so now it can be recommended for publication